# Functional and Proteomic Insights into Aculeata Venoms

**DOI:** 10.3390/toxins15030224

**Published:** 2023-03-16

**Authors:** Daniel Dashevsky, Kate Baumann, Eivind A. B. Undheim, Amanda Nouwens, Maria P. Ikonomopoulou, Justin O. Schmidt, Lilin Ge, Hang Fai Kwok, Juanita Rodriguez, Bryan G. Fry

**Affiliations:** 1Australian National Insect Collection, Commonwealth Scientific & Industrial Research Organisation, Canberra, ACT 2601, Australia; 2Venom Evolution Lab, School of Biological Sciences, The University of Queensland, St. Lucia, QLD 4072, Australia; 3Centre for Ecological and Evolutionary Synthesis, Department of Bioscience, University of Oslo, N-0316 Oslo, Norway; 4School of Chemistry and Molecular Biosciences, University of Queensland, St. Lucia, QLD 4072, Australia; 5Translational Venomics Group, Madrid Institute for Advanced Studies in Food, 4075 Madrid, Spain; 6Southwestern Biological Institute, 1961 W. Brichta Dr., Tucson, AZ 85745, USA; 7State Key Laboratory Cultivation Base for TCM Quality and Efficacy, School of Pharmacy, Nanjing University of Chinese Medicine, 138 Xianlin Avenue, Qixia District, Nanjing 210046, China; 8Institute of Translational Medicine, Department of Biomedical Sciences, Faculty of Health Sciences, University of Macau, Avenida da Universidade, Taipa, Macau

**Keywords:** Aculeata, venom, sociality, proteomics, cytotoxicity

## Abstract

Aculeate hymenopterans use their venom for a variety of different purposes. The venom of solitary aculeates paralyze and preserve prey without killing it, whereas social aculeates utilize their venom in defence of their colony. These distinct applications of venom suggest that its components and their functions are also likely to differ. This study investigates a range of solitary and social species across Aculeata. We combined electrophoretic, mass spectrometric, and transcriptomic techniques to characterize the compositions of venoms from an incredibly diverse taxon. In addition, in vitro assays shed light on their biological activities. Although there were many common components identified in the venoms of species with different social behavior, there were also significant variations in the presence and activity of enzymes such as phospholipase A_2_s and serine proteases and the cytotoxicity of the venoms. Social aculeate venom showed higher presence of peptides that cause damage and pain in victims. The venom-gland transcriptome from the European honeybee (*Apis mellifera*) contained highly conserved toxins which match those identified by previous investigations. In contrast, venoms from less-studied taxa returned limited results from our proteomic databases, suggesting that they contain unique toxins.

## 1. Introduction

The order Hymenoptera is hyperdiverse and contains a significant plurality—perhaps even a majority—of all extant venomous species [1,2,3,4,5]. These insects play a major role in almost every terrestrial ecosystem but are also significant in terms of purely human concerns because of their capabilities as pests [6,7,8,9], biocontrol agents [10,11,12], agricultural pollinators [13,14], and even threats to human life [15,16,17,18].

Within Hymenoptera, the subclade Aculeata could be said to contain the most diverse array of life histories and social behaviors (including predatory, parasitic, and pollinivorous taxa). Eusociality has arisen multiple times in insects and is another major axis of variation among aculeate lifestyles [19]. Many aspects of sociality—including the underlying genetic systems and selection pressures which lead to it [20,21], and its consequences on life span, resistance, and senescence—in eusocial and solitary species [22], have been studied. Since venom composition often correlates with the behavior of the organism, it would seem likely that venom composition would also change with this evolutionary transition. Solitary and parasitic aculeate wasps use their venoms in order to paralyze and preserve their prey [23,24,25,26], whereas the venoms of bees (both solitary and social) and other social aculeates are primarily deployed in defense of themselves or their colonies [27,28,29,30]. A review of the toxins found in Vespid venoms concluded that the social and solitary species of that family express very different toxins from each other [31]. However, it remains unclear whether social lifestyles have had similar effects on the composition and biochemical activities of the venoms from the various aculeate lineages which have independently evolved towards eusociality. To begin to understand this phenomenon, it is necessary to compare venom composition and activity in a wide range of species.

Aculeate venoms are mixtures of peptides, enzymes, biogenic amines, and other organic compounds, such as formic acid [32,33,34,35,36,37]. Despite solitary species measurably outnumbering their social counterparts [38,39,40], the majority of venom research has focused on social species, in particular, the honeybee, *Apis mellifera* [1]. The venom from species of Vespidae and Formicidae have also received attention, mostly due to their ability to cause allergic reactions in humans [28,31,41,42]. Sensitivity to these venoms can arise through IgE-mediated, non-IgE-mediated, or even nonimmunologic mechanisms; and more than ten (enzymatic and non-enzymatic) allergens from *A. mellifera* venom alone have been studied [43,44]. However, most research has focused on the characterization and isolation of single molecules [22,45,46,47,48,49,50,51,52], thereby neglecting whole venom composition. This approach can occasionally overlook evolutionarily relevant findings in cases where an important venom function arises from the interaction or synergy between different toxins.

Many aculeate venoms cause generalized pain and inflammation, and occasionally, they cause anaphylactic shock [53]. A recent review suggests that Phospholipase A_2_s (PLA_2_s) are likely the main allergenic component of *A. mellifera* venom, but other toxins, including serine protease enzymes and hyaluronidases, account for much of the allergenicity as well [54]. Similar toxins have been found in both solitary and social venoms [31,55]; however, in many species, their functions have been implied rather than experimentally tested. Recent studies have identified small linear peptidic toxins from a range of aculeates that also disrupt cell membranes by forming amphipathic helices [56,57,58,59,60,61,62]. Experimentally investigating these bioactivities will give a better understanding of species-specific venom activity. Damage caused by aculeate venoms is often the result of cytotoxic components. Such components have been reported to have potential anticancer effects, which have been extensively explored in bee venom [63,64,65,66,67] but neglected in the majority of other aculeate species, with few exceptions [58,68,69,70]. Further exploring the cytotoxic abilities of venoms will be instrumental guiding translational research exploring the potential to create anti-cancer drugs inspired by these venoms. Other toxins may also be involved in similar lines of research investigating possible anti-inflammatory medications [71,72,73,74,75,76,77].

Modern technologies, especially transcriptomic and proteomic techniques, have made it easier to begin to unravel the compositions of whole venoms. This study involved a proteo-transcriptomic analysis of *Apis mellifera* venom and large-scale comparisons of aculeate venom using a variety of -omic and bioactivity analyses to increase our understanding of these venoms and the broad patterns of venom variation in these insects (Figure 1).

## 2. Results

### 2.1. Transcriptome

After quality control and annotation, our *Apis mellifera* venom gland libraries yielded transcripts whose translated sequences are almost identical to the protein sequences of previously reported *A. mellifera* toxins (Figure 2). These final toxin transcripts included icarapins, phospholipase A_2_ (PLA_2_), anthophilins including apamin [78], and carboxylesterases.

### 2.2. Proteomics

1D SDS-PAGE results suggested only small variances in the molecular masses or toxins between species of the same genus, but much greater differences among genera (Figure 3 and Figure 4).

### 2.3. LC-MS

Venoms were also profiled using LC-MS to examine the low-molecular-mass components. All venoms showed a similar generalized elution profile, revealing venoms rich in low-molecular-mass components (Figure 5, Figure 6 and Figure 7). The components were distributed over the molecular mass range of 500–14,000 Da. The lack of high-mass toxins in the chromatographs does not indicate a true absence. It is more likely a result of ion suppression, which is common in LC-MS analyses [80]. Social bee venoms showed similar chromatograms with evidence of peptide variability among species. However, the chromatograms of solitary bee venoms had distinctly fewer peaks, despite their relatively rich proteomic profiles (Figure 3B and Figure 5F). Wasp venom composition showed significant similarities across species in retention times and molecular masses (Figure 6A–D), as did the ants (Figure 7).

### 2.4. LC-MS/MS

Despite the high diversity of toxins shown to be present in the gels (Figure 3 and Figure 4) and LC-MS chromatographs (Figure 5, Figure 6 and Figure 7), shotgun-MS/MS analysis was only able to find similar matches to a relative handful of toxins (Figure 8). This was especially pronounced in the solitary wasps and was likely because there are relatively few published homologous sequences available in public databases for us to search our mass spectra against.

### 2.5. Enzymatic Assays

High PLA_2_ activity was found in all social bee venoms (Figure 9) compared to the rest of Aculeata. Statistical investigations provided support for social species being more likely to have higher PLA_2_ activity (PGLS: t = 3.27, df = 1, p = 0.002). However, when looking at the cleavage of serine protease specific substrate, some of the solitary bees, including *Xylocopa rufa*, *X. californica*, and *Peponapis pruinosa*, were the most active, alongside some of the *Polistes* species (Figure 9).

### 2.6. Cytotoxicity Assays

The cytotoxic effects of whole venom on one non-transformed and one cancerous cell line were tested to ascertain generalized cytotoxicity (Figure 10). The results showed that the majority of social bee venoms had strong cytotoxic tendencies against both cell lines, as did ant venoms (particularly the genus *Mymercia*). Using statistical measures, we found that the high cytotoxicity against both non-transformed and cancerous cell lines was related to social aculeates: MM96L (PGLS: t = 3.22, df = 1, p = 0.002); NFF (PGLS: t = 2.87, df = 1, p = 0.005). Further, this higher cytotoxicity against the non-transformed and cancerous cell lines was also statistically significant (PGLS: t = 10.92, df = 1, p = 2 × 10^−16^).

## 3. Discussion

In order to fully characterize the venoms of aculeates, a comparative study of venomgland transcriptomes and proteomes is necessary. In recent years, the number of studies that included these data for hymenopterans has increased, but in the face of the enormous diversity of the order, it is clear that the research community has only started to scratch the surface of what there is to be discovered [62,78,91,92,93,94,95,96,97,98,99,100,101,102,103,104]. One interesting aspect of our own contribution to this enormous task is that the transcripts we identified from the venom gland of *A. mellifera* were found to have nearly identical sequences to other *A. mellifera* venom proteins which are available in the Uniprot database (Figure 2) and those identified by Koludarov et al. [78]. This similarity could be due to reduced genetic diversity in this species (perhaps as a result of domestication), or it could indicate an unusually strong pattern of conservation in these genes. Studies of honeybees’ genetic diversity suggest that there have been some declines, but that diversity remains reasonably high in this species [105,106,107,108,109,110]. Therefore, a lack of underlying genetic diversity is unlikely to account for the extreme conservation observed in these toxins. Defensive venoms have frequently been noted to be less variable than predatory venoms, so the purpose of the venom may help explain the extreme similarity of *A. mellifera* toxins [111,112,113]. More specifically, this accords with the finding of Koludarov et al. [78] that the core hymenopteran venom genes are strongly conserved throughout the evolutionary history of the order. Despite the identification of most of the major venom toxins, some of the previously described venom compounds were not able to be recovered. One of these was the antigen 5-like wasp venom paralog, which was absent from the venom gland’s transcriptome. This venom protein is known to be seasonally expressed, and this may have been the reason for its absence in the transcriptome [114].

We also presented a broad functional overview of the venom of aculeate species from the major Aculeata clades that include solitary and social species: Vespidae, Formicidae, and Apoidea; and from two clades with solitary species only: Mutillidae and Scoliidae. Proteomic analysis consisting of 1D SDS-PAGE and LC-MS, combined with shotgun-MS/MS, revealed a diversity of toxins present in both solitary and social species. The 1D SDS-PAGE and LC-MS results suggest a lack of systematic differences between the venoms of social and solitary hymenopterans. Moreover, very similar profiles were observed between congeneric samples (Figure 3, Figure 4, Figure 5, Figure 6 and Figure 7). Despite the diversity of peptides revealed by these methods, a negligible number of toxins in solitary species were similar enough to any reference toxins to produce a hit using shotgun-MS/MS (Figure 8). Previous studies have found solitary wasp venoms to mostly be rich in proteins that are used in order to kill and immobilize prey [26,31,93,94,115,116], whereas solitary bee venoms often contain more antimicrobial peptides [45,46,47,48,49,104,117,118,119]. Previous research has proposed the existence of a hyperdiverse family of peptides from aculeate venoms known as aculeatoxins which tend to form amphipathic helices [62]. The extreme variability in the sequences of the mature toxins from this family would make it difficult to detect novel members using a library-based approach such as shotgun-MS/MS. Overall, the small number of peptides identified from solitary species is unlikely to stem from a genuine absence of toxins but is probably a result of limitations of the proteomics reference database. There are scant sequences available for use as reference material from much of the hymenopteran phylogeny.

LC-MS results revealed an abundance of low-mass molecules (Figure 5, Figure 6 and Figure 7), which is consistent with previous studies suggesting the prevalence of biogenic amines in bee and wasp venoms [120], alkaloids in ant venoms [42], and widespread abundance of small peptides, including allergens across the order [1,33]. Sequence similarities between some of these small linear peptidic toxins—especially in the signal peptide region—formed the basis for the aculeatoxin hypothesis, which suggests that these toxins are related to one another and form a toxin superfamily [62].

PLA_2_s and serine proteases can be significant allergens in aculeate venoms [32,33,121]. PLA_2_s are known to be the main enzymes found in honeybee venoms, making up approximately 12% of the dry mass of venom [122,123]. Comparatively, wasp venoms have been found to only have 0.1–1% of the protein present [51], and ants have been found to have similarly low levels of PLA_2_s [124]. Concordantly, our results indicate that social bee venoms have higher levels of PLA_2_ activity than most other hymenopterans, and elevated activity in the venoms of *Xylocopa rufa* and *Tetraponera* sp. as well (Figure 9). This suggests that toxins other than PLA_2_s are more likely to be responsible for allergic reactions to the venoms of other taxa. This pattern is quite different to that of serine protease activity, which was elevated in some but not all species of *Xylocopa* and *Polistes*, and in *Peponapis pruinosa* (Figure 9). The molecular function of serine proteases in bee venom is still unknown.

Cytotoxicity is another well-documented activity in aculeate venoms [58,64,68,125], and it has been hypothesized to serve the defensive function of inducing pain. Our results show that the venoms of social bees and some ants—particularly the subfamily Myrmecinae—are more cytotoxic than other hymenopterans, and this pattern was accentuated in the cancerous MM96L cells compared to the non-cancerous NFF cell. This pattern suggests that cytotoxicity has indeed evolved independently in some social lineages, potentially for the purpose of colony defense. The cytotoxic effects of the venom of *Apis* species is mainly due to the peptide melittin via a membranolytic effect [126]; PLA_2_s have also been shown to synergistically increase melittin’s cytotoxic effects [73]. The fact that melittin is not present in solitary bee venom suggests that it is likely the primary driver of cytotoxicity in social bee venom. Pilosulin and other aculeatoxins from the *Myrmecia* genus have been identified as potently cytotoxic molecules [58,62]. Cytotoxic molecules that have been identified in social wasps include mastoparan, which targets the mitochondrial membrane, resulting in tumor cell cytotoxicity [127], and a biologically active quinone isolated from *Vespa simillima* venom which induces apoptosis [128]. Mastoparans have been isolated from solitary Vespidae but no other species of solitary wasps [31], perhaps hinting at their predominant role in causing the cytotoxic effects of these species. The use of venom peptides for cancer-specific drugs is not a new idea, but no lead compound from a venom has led to an approved anti-cancer drug for human use so far. This is mostly due to the difficulty in isolating peptides that are able to discriminate between deleterious cells and healthy cells. In this study, we found that a range of aculeate venoms are cytotoxic. While many of them were equally damaging to both cell lines, some venoms—especially those of several *Myrmecia* species—were notably more toxic to cancerous cells than non-cancerous. Other studies have reported that specific peptides from aculeate venoms have various anti-cancer and anti-tumour activities and thus are good potential candidates for these therapeutic avenues [63,126,129,130], and our results suggest some further targets for this sort of in-depth research.

While PGLS analyses indicate that venoms from social lineages display statistically higher PLA_2_ activity and cytotoxicity (see Section 2.5 and Section 2.6), these are due to elevated levels in the social bees in the first instance and in social bees and some ants in the second. There is no single activity that shows a strong sign of being upregulated in social species from all clades and low levels in the solitary species. This suggests that, while sociality clearly alters the selection pressures acting upon the venoms of these lineages, it does not favor any one particular solution, and the actual toxins and mechanisms employed by social and solitary hymenopteran venoms are often lineage-specific.

## 4. Conclusions

This study offered a broad investigation into venoms from aculeate hymenopterans to help understand their compositions, functions, and evolution. We also sequenced the venom-gland transcriptome of a honeybee, *A. mellifera*, which showed that the toxins are extremely conserved across the species. Venom fingerprinting with 1D-SDS PAGE gels and LC/MS suggests that venom composition is often similar within genera, but can vary greatly even between closely related genera. Proteomics and mass spectrometry studies revealed these venoms include a diversity of small peptides, but most were not able to be identified. This suggests that they bear little resemblance to previously discovered toxins which we were able to include in our reference databases.

Our PLA_2_ activity and cytotoxicity assays suggested significant differences between the venoms of social and solitary species. In each of our assays, these results were driven by particular groups of social aculeates—social bees showed high levels of PLA_2_ activity and cytotoxicity, social wasps had elevated serine protease activity, and ants possessed all of the most cytotoxic venoms we tested—rather than identifying a particular toxin family or mechanism that showed a clear difference between all social and all solitary species. That being said, these components are mainly pain and/or damage-inducing, and the social lineages responsible for these significant signals upregulated these activities. This suggests that the venoms of social species may have independently evolved to ward off predators but that each lineage achieves this goal using different toxins.

These results add to a growing body of evidence suggesting that hymenopteran venoms have a somewhat paradoxical nature. Many of the venom toxins are highly conserved throughout the entire evolutionary history of the order [78], but others are so diverse that they cannot be identified from mass spectra without highly-related reference sequences to compare against. Many aculeate venoms serve highly similar functions (e.g., defensive venoms in social taxa), but they appear to carry out these roles by employing different toxins and biochemical mechanisms. Findings of incredibly conserved core venom genes or strong negative selection on toxin sequences might be taken to mean that investigating a handful of aculeate venoms would tell us most of what there is to know about the venoms from other members of the clade, but our results suggest that there remains an incredible diversity of toxins and mechanisms to be discovered in the venoms of unstudied aculeate taxa. This diversity will prove useful to future researchers interested in the lives and ecology of these insects and provides a wealth of leads for those looking for new investigational ligands or scaffolds for drug design and development in animal venoms.

## 5. Materials and Methods

### 5.1. Taxonomic Selection

The species included in this study (Table 1) were selected in order to provide phylogenetically diverse coverage of aculeate clades that have both solitary and social species (Apoidea, Vespidae, Formicidae) and some that are purely solitary (Mutillidae, Scoliidae).

### 5.2. Venom Collection

For most species, the venom reservoirs were dissected from the body, rinsed in distilled water, and torn open to let the venom drain out. The venom was then collected for study and the empty reservoir discarded. However, social wasp venoms were collected as described by Schmidt et al. [131]: the sting apparatus was pulled from the body of cold anesthetized wasps, and then, the muscular venom sac was gently squeezed while holding the sting tip to fine Dumont #5 forceps. This expressed the venom which flowed through the stinger and by capillary action up the tines of the forceps.

Venoms were pooled from multiple individuals for each sample, and the number of individuals varied based on venom yield and the ability to collect specimens.

### 5.3. Proteomics

#### 5.3.1. SDS-PAGE

One-dimensional (1D) sodium dodecyl sulfate polyacrylamide gel electrophoresis (SDS-PAGE) was carried out as previously described [132,133,134]. Twelve-percent SDS-PAGE gels were cast into 1 mm slabs with a resolving gel layer (3.3 mL Milli-Q H2O, 4 mL 30% acrylamide mix, 2.5 mL 1.5 M Tris–HCl buffer, pH 8.8, 100 μL 10% SDS, 4 μL TEMED, 100 μL 10% APS); 20 μg venom sample per lane after dissolving in 3 μL of 4× sample loading buffer (12 μL total volume) with DTT; reducing conditions were 3 min incubation at 100 °C; gels were run at room temperature at 120 V for 20 min and then 140 V for 60 min; runs were stopped when dye front was less than 10 mm from the base of the gel (Mini Protean3, Bio-Rad Lab). Gels were stained with colloidal Coomassie brilliant blue G250 (34% methanol, 3% phosphoric acid, 170 g/L ammonium sulphate, 1 g/L Coomassie blue G250) overnight and then destained in 1% acetic acid.

#### 5.3.2. Liquid Chromatography–Mass Spectrometry (LC-MS)

LC-MS and HPLC analyses of 25 μg crude venom was performed on a Nexera system (Shimadzu: Kyoto, Japan) using a Zorbax 300SB C18, 3.5 μm column (2.1 × 100 mM, Agilent) at a flow rate of 300 μL/min. The gradient used was 2–40% Buffer B (90% acetonitrile) over 35 min, 40–98% Buffer B for 2 min, and then holding at 98% Buffer B for 2 min. Buffer A was 0.1% formic acid in water. The HPLC was directly connected to a 5600 TripleTOF equipped with a DuoSpray™ ion source (SCIEX, Framingham, MA, USA), operated in positive-ion acquisition mode. Data were acquired for 46 min over the *m*/*z* range 350–2000 Da with a cycle time of 0.5 s. Raw results were analyzed in Analyst® (SCIEX, Framingham, MA, USA).

#### 5.3.3. Tandem Mass Spectrometry (LC-MS/MS)

For liquid chromatography–tandem MS (LC-MS/MS), venom was centrifuged (10 min, 12,000 rcf, 4 °C) to remove particulate matter, and 5–50 μg of clarified venom was incubated with 20 μL reduction/alkylation buffer (50 mM ammonium carbonate pH 11.0, 1% iodoethanol, 0.025% triethylphosphine in 48.5% acetonitrile) for 2 h at 37 °C. The reduced and alkylated sample was then lyophilized and resuspended in 10 μL digestion reagent (20 ng/μL proteomics grade trypsin Sigma #T7575, in 40 mM ammonium bicarbonate pH 8.0, 5% acetonitrile) for 16 h at 37 °C. The reaction was then terminated by addition of 20 μL 5% formic acid, and the tryptic digest was lyophilized. Digests were resuspended in 1% formic acid and 2.5% acetonitrile and loaded onto a 150 × 0.1 mm Zorbax 300SB-C18 column (3.5 μm particle size, 300 Å pore size, Agilent catalog no. 5065-9910) on a Shimadzu Nano LC system. The LC outflow was coupled to a SCIEX 5600 Triple TOF mass spectrometer equipped with a Turbo V ion source. Peptides were eluted over a 70 min gradient of 1–40% solvent B (90% acetonitrile, 0.1% formic acid) in solvent A (0.1% formic acid) at a flow rate of 0.2 mL/min. MS1 scans were collected between 350 and 1800 *m*/*z*, and precursor ions in the range *m*/*z* 350–1500 with charge +2 to +5 and signal >100 counts/s were selected for analysis, excluding isotopes within 2 Da. MS/MS scans were acquired with an accumulation time of 250 ms and a cycle time of 4 s. The "rolling collision energy" option was selected, allowing collision energy to be varied dynamically based on *m*/*z* and z of the precursor ion. Up to 20 similar MS/MS spectra were pooled from precursor ions, differing by less than 0.1 Da. The resulting mass spectra in WIFF format were then compared with a library of translated ORFs extracted from transcriptomes generated from RNA-Seq experiments (together with a list of common MS contaminants) using a Paragon 4.0.0.0 algorithm implemented in ProteinPilot 4.0.8085 software (SCIEX). A mass tolerance of 50 mDa was used for both precursor and MS/MS ions.

### 5.4. Transcriptomics

#### 5.4.1. RNA Extraction and Library Preparation

Ten female *Apis mellifera* were collected from EcoSciences Precinct, University of Queensland, Australia. The venom glands were isolated by dissection, and total RNA was extracted from venom glands by standard TRIzol protocol (ThermoFisher, Waltham, MA, USA). The RNA sample was submitted to the University of Queensland Institute for Molecular Bioscience Sequencing Facility for library preparation and sequencing. A paired-end library with 180 bp insert size was constructed using the Illumina TruSeq-3 Stranded mRNA kit and sequenced on an Illumina NextSeq using a 300-cycle (2 × 150 bp) mid-output run. These reads are available at SRA SRR11349374.

#### 5.4.2. Sequence Data Pre-Processing and Transcriptome Assembly

The resulting reads were trimmed using Trimmomatic v0.35 [135] to remove adapter sequences and low-quality reads. Window-function-based quality trimming was performed using a window size of 4 and a window quality of 20, and sequences with a resulting length of <100 bp after trimming were removed. The trimmed reads were de novo assembled into contigs by Trinity v2.4.0 [136] using default parameters.

#### 5.4.3. Transcriptome Annotation

The de novo assemblies were concatenated and searched against reference toxin sequences obtained from UniProt using BLAST version 2.7.1 [137,138]. CD-HIT v4.7 was used to cluster the sequences and remove duplicates [139,140]. The remaining contigs that did not contain complete coding sequences were removed. Final toxin sequences were visualized and aligned to homologues from the Uniprot database using AliView v1.26 [79]. Annotated CDS sequences are available on GenBank under the accession numbers OM416840-OM416850 in the BioProject PRJNA613391.

### 5.5. Bioactivity Activity Testing

#### 5.5.1. Enzymatic Activity Studies

A Thermo Scientific™ Fluoroskan Ascent™ Microplate Fluorometer was employed to test variation in enzymatic activity. A fluorescence substrate assay (E10217 EnzChek® Phospholipase A_2_ Assay Kit, ThermoFisher Scientific) was used for assessing the PLA_2_ activity. Venom solution (0.1 μg in dry venom mass) was brought up to 12.5 μL in PLA_2_ reaction buffer (250 mM Tris–HCL, 500 mM NaCl, 5 mM CaCl_2_, pH 8.9) and plated out in triplicate on a 384 well plate. Triplicates were measured by adding 12.5 μL quenched 1 mM EnzChek® Phospholipase A_2_ substrate per well (total volume 25 μL/well) over 100 cycles at an excitation of 485 nm and emission of 520 nm, using a Fluoroskan Ascent (ThermoFisher Scientific). The negative control consisted of PLA_2_ reaction buffer and substrate only.

For testing on Mca-PLGL-Dpa-AR-NH2 fluorogenic peptide substrate (Cat. # ES001, R&D systems, Minneapolis, Minnesota), 10 μL of 0.05 μg/μL venom stock was plated in triplicate on a 384-well black plate and measured by adding 90 μL quenched fluorescent substrate per well. The substrate concentration of each substrate stock solution dissolved into 4.990 mL of enzyme buffer (150 mM NaCl and 50 mM Tris-HCl pH 7.4) was 10 μL. Fluorescence was monitored over 400 min or until activity ceased. Excitation was at 390 nm and emission was at 460 nm for substrate ES011. The machine was programmed to shake the plate for three cbefore each reading to maintain homogeneity in the wells. Relative enzymatic activity was calculated as an increase in absorbance corresponding to the cleavage of the fluorescent group. Finally, the raw data were normalized to meet analysis assumptions and processed with GraphPad Prism 7.0.

#### 5.5.2. Cytotoxicity Studies

The effect of each venom was assessed on human neonatal foreskin fibroblast (NFF) and malignant melanoma (MM96L) cell lines, supplied by QIMR Berghofer Medical Research institute. Venom-mediated cytotoxicity is often responsible for the degradation and destruction of skin and connective tissue. Therefore, the chosen cell lines were deemed appropriate. Cell lines were maintained in RPMI medium supplemented with 1% penicillin streptomycin and fetal calf serum (FCS), 10% FCS for NFF, and 5% FCS for MM96L. FCS was heat inactivated at 56 °C for 20 min. Endotoxin was tested and accepted if ≤10 EU/mL. Cells were split 24 h prior to the experiment (for up to 25 passages for MM96L and 10 passages for NFF) using 0.25% trypsin and seeded in 96 well flat-bottom plates at a density of 5000 and 2500 cells/well for NFF and MM96L cells, respectively. Trypan blue was used to accurately seed and plate an equal number of cells per treatment. Plates were incubated overnight at 37 °C in a 5% CO_2_ 95% humidified environment prior to treatment. Cell viability was evaluated using colorimetric MTT (Thiazolyl Blue Tetrazolium Bromide; Sigma Aldrich M5655, Sydney, NSW, Australia) assays. Venom was added to cells at 5 μg and 0.5 μg protein amounts and followed by a 48 h incubation period. MTT was added at a concentration of 5 mg/mL per well. An amount of 0.1% sodium dodecyl sulfate (SDS) was used as a positive control to achieve 100% toxicity, and the protocol was followed according to the manufacturer’s description. The absorbance was read at 570 nm on the PowerWave XS2 plate reader (Bio Tek Instruments, Winooski, VT, USA), using Gen5 software. Two independent experiments were conducted with a minimum of three replicates per treatment. Cell viability readings were normalized as percentages of untreated control cells, and viability is expressed as a percentage of toxicity ± standard error of the mean (SEM). The relationship between venom dose and cytotoxic response was calculated via area under the curve (AUC) analysis, using GraphPad Prism 7 (GraphPad Software, Inc., La Jolla, CA, USA).

### 5.6. Ancestral State Reconstruction

No single published phylogeny included all the species in our sample, so the topology and branch lengths were manually assembled using a variety of different sources. TimeTree was able to provide time-calibrated phylogenies for some species and subclades and references to the original studies [83]. Other taxa and dates were added using data from a range of previously published phylogenies [81,82,84,85,86,87,88,89]. The phylogeny was built and edited using Mesquite 3.7 [90].

The resulting phylogeny was imported into the statistical software R (version 3.6.1) using the APE package [141]. The contMAP function of the phytools package was used to estimate ancestral states, using maximum likelihood, and to visually represent the presented trait over the tree [142]. Four trees were produced: two for the enzymatic assays measuring PLA_2_ and serine protease activity, and two for the assays measuring cytotoxicity in melanoma and NFF cells. This protocol has been described previously [143].

## Figures and Tables

**Figure 1 toxins-15-00224-f001:**
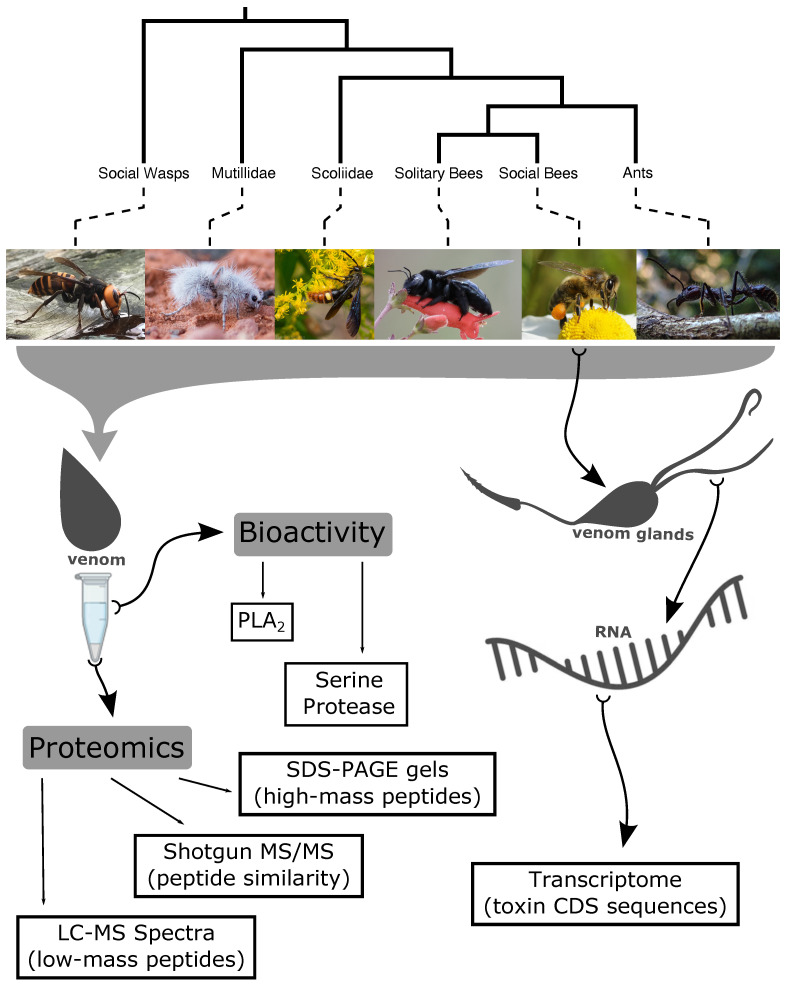
Schematic overview of the key aculeate groups sampled in this study, the samples derived from them, and the data generated. Photo species and credit (left to right): *Vespa mandarinia* (Asian giant hornet) by Gregory Mihaich under CC-BY-NC-SA, *Dasymutilla gloriosa* (thistledown velvet ant) by mrwood under CC-BY-NC, *Scolia dubia* (blue-winged flower wasp) by Thomas Shahan under CC-BY-NC, *Xylocopa californica* (western carpenter bee) Arman Moreno under CC-BY-NC, *Apis mellifera* (honeybee) Sandy Rae under CC-BY-SA, *Paraponera clavata* (bullet ant) by manimiranda under CC-BY-NC. All images were retrieved from iNaturalist (https://www.inaturalist.org/).

**Figure 2 toxins-15-00224-f002:**
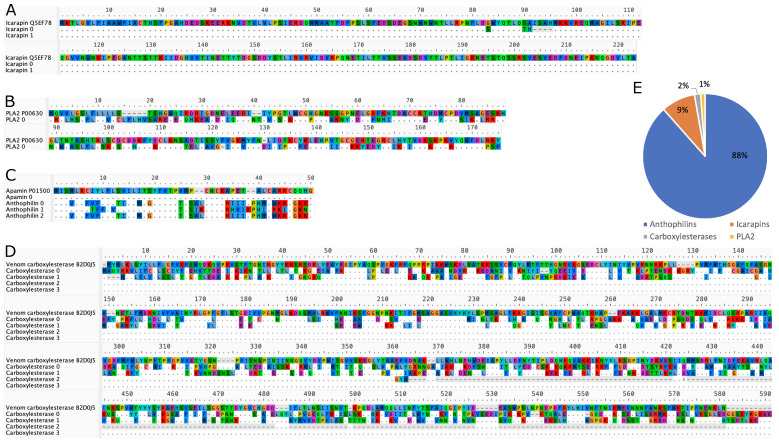
(**A**–**D**) Alignment of translated toxin CDS sequences with the sites with UniProt references. Residues identical to the reference are replaced by 
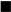
, and amino acids are colored according to the default settings of AliView [79]. Toxin families include: (**A**) icarapins, (**B**) phospholipase A_2_, (**C**) anthophilins such as apamin [78], (**D**) carboxylesterases. (**E**) Relative length-normalized expression of these toxin families in the transcriptome, measured as total RPK for each family.

**Figure 3 toxins-15-00224-f003:**
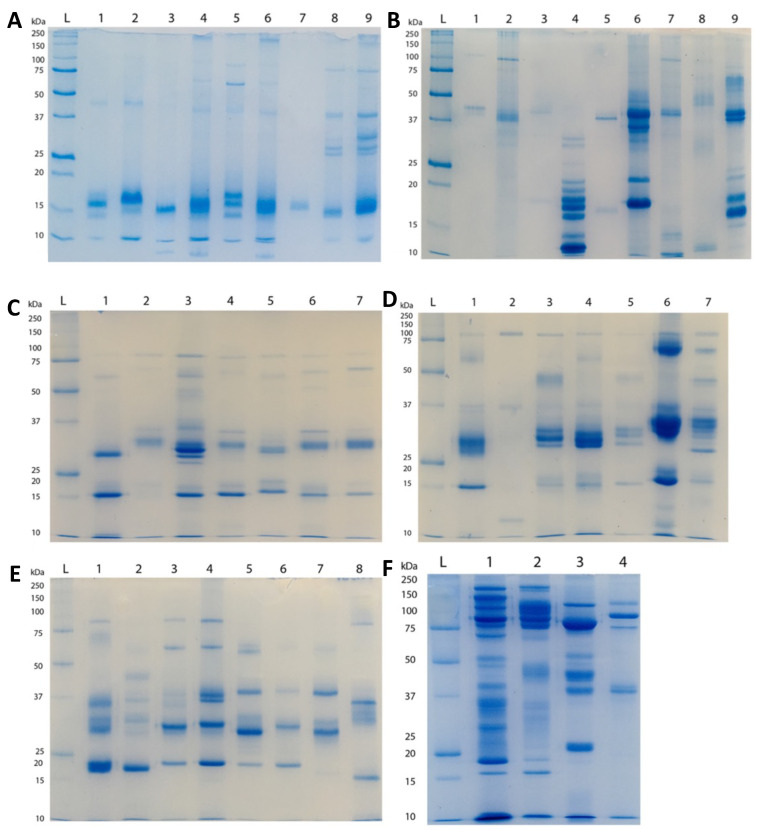
1D SDS-PAGE (12% acrylamide with Coomassie brilliant blue staining) of venom from bees and wasps: (**A**) social bees (reduced); 1 = *Apis mellifera* (European); 2 = *A. mellifera* (Africanised); 3 = *A. andreniformis*; 4 = *A. cerana*; 5 = *A. dorsata*; 6 = *A. florea*; 7 = *A. koschevnikovi*; 8 = *Bombus huntii*; 9 = *B. impatiens*. (**B**) Solitary bees (reduced); 1 = *Centris aethycetra*; 2 = *C. rhodipus*; 3 = *Diadasia rinconis*; 4 = *Peponapis pruinosa*; 5 = *Xylocopa rufa*; 6 = *X. californica*; 7 = *Crawfordapis* sp.; 8 = *Lasioglossum kinabalueuse*; 9 = *X. veripuncta*. (**C**) Epiponini wasps (reduced); 1 = *Agelaia myrmecophila*; 2 = *Brachygastra mellifica*; 3 = *Polistes flavus*; 4 = *Polybia rejecta*; 5 = *Polybia sericea*; 6 = *Polybia simillima*; 7 = *Synoeca septentrionalis*. (**D**) *Polistes*, Ropalidini, and Mischocyttarini wasps (reduced); 1 = *Belonogaser juncea colonialis*; 2 = *Mischocyttarus flavitarsus*; 3 = *Polistes canadensis*; 4 = *Polistes comanchus navajoe*; 5 = *Polistes dorsalis*; 6 = *Parachartergus fraternus*; 7 = *Polistes major castaneocolor*. (**E**) Vespinae wasps (reduced); 1 = *Dolichovespula arenaria*; 2 = *D. maculata*; 3 = *Vespula pensylvanica*; 4 = *Vespula vulgaris*; 5 = *Vespa luctuosa*; 6 = *Vespa simillima*; 7 = *Vespa tropica*. (**F**) Solitary wasps (reduced); 1 = *Dasymutilla chiron*; 2 = *D. gloriosa*; 3 = Scoliidae; 4 = *Stictia*.

**Figure 4 toxins-15-00224-f004:**
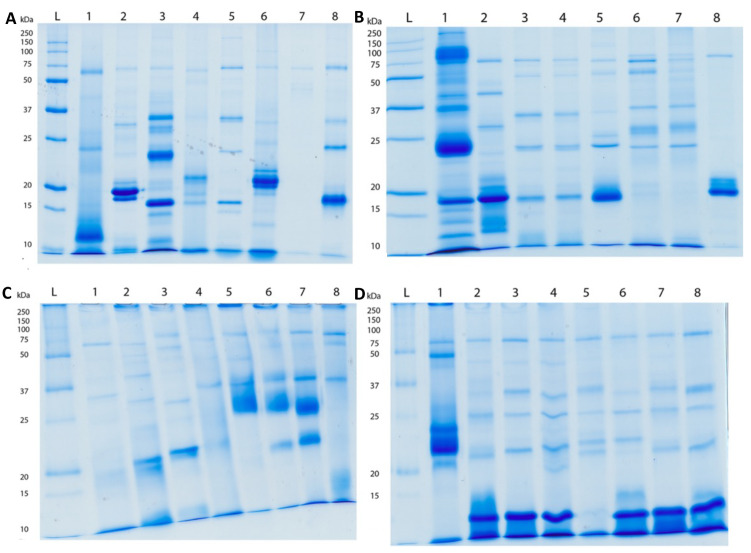
1D SDS-PAGE (12% acrylamide with Coomassie brilliant blue staining) of venom from ants (reduced): (**A**) 1 = *Paraponera clavata*; 2 = *Diacamma*; 3 = *Euponera sennaaren*; 4 = *Leptogenys*; 5 = *Neoponera villosa*; 6 = *Odontomachus*; 7 = *Opthalmopone*; 8 = *Megaponera analis*. (**B**) 1 = *Pachycondyla crassinoda*; 2 = *Paltothyreus tarsatus*; 3 = *Platythyrea lamellosa*; 4 = *P. strigulosa*; 5 = *Streblognathus aethiopicus*; 6 = *Neoponera commutata*; 7 = *N. commutata* (Queen); 8 = *Odontoponera*. (**C**) 1 = *Ectatomma tuberculatum*; 2 = *Ectatomma*; 3 = *Gnaptogenys*; 4 = *Rhytidoponera metallica*; 5 = *Pogonomyrmex maricopa*; 6 = *P. occidentalis*; 7 = *P. rugosus*; 8 = *Diacamma*. (**D**) 1 = *Tetraponera* sp.; 2 = *Myrmecia browningii*; 3 = *M. gulosa*; 4 = *M. nigripes*; 5 = *M. pilosula*; 6 = *M. rufinodis*; 7 = *M. simillima*; 8 = *M. tarsata*.

**Figure 5 toxins-15-00224-f005:**
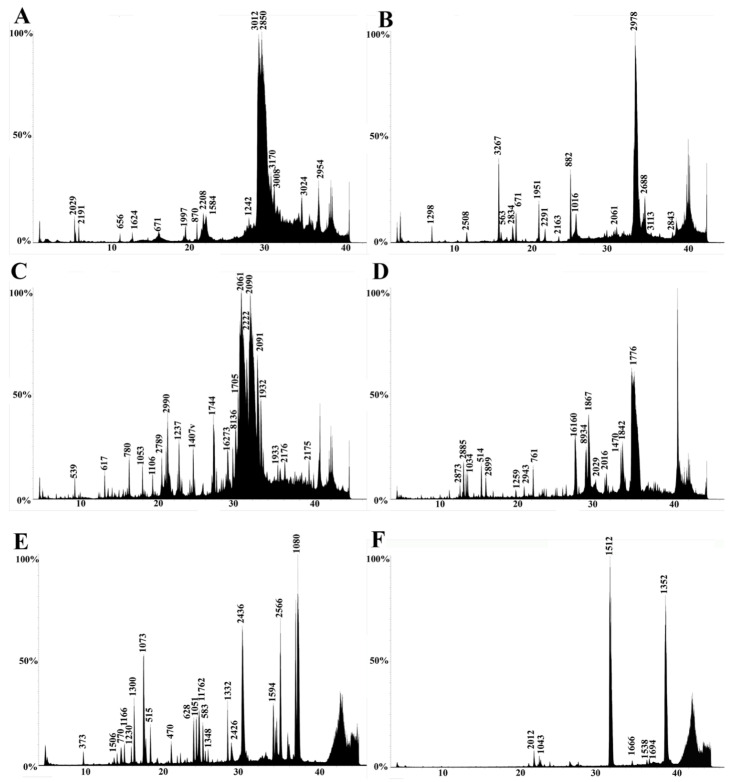
Representative LC-MS profiles of bee species: (**A**) *Apis mellifera*, (**B**) *A. andreniformis*, (**C**) *Bombus impatiens*, (**D**) *B. sonorus*, (**E**) *Xylocopa californica*, (**F**) *Peponapis pruinosa*. The x-axis is time (minutes); the y-axis is relative intensity (0–100%). Reconstructed mass in Daltons is shown above each peak.

**Figure 6 toxins-15-00224-f006:**
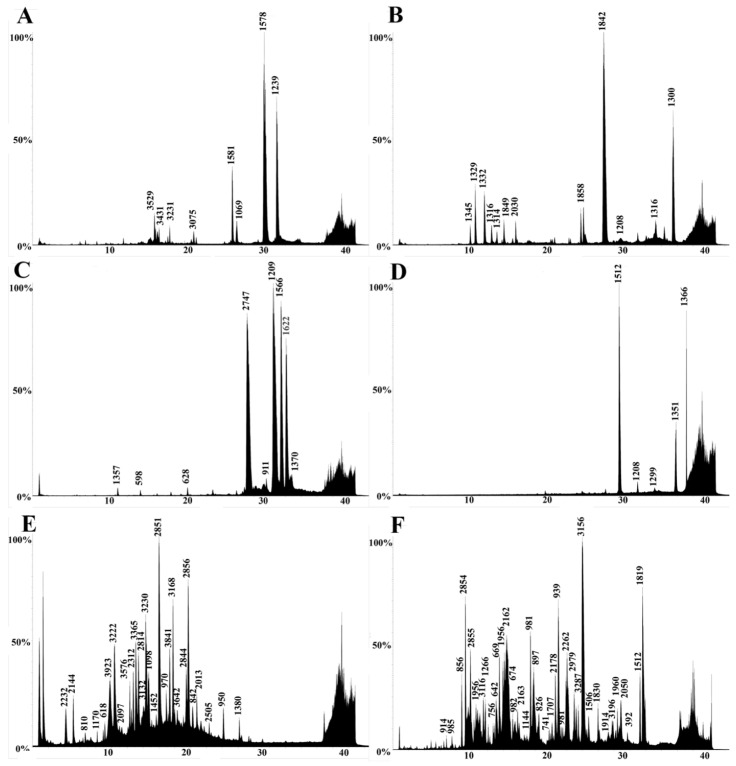
Representative LC-MS profiles of wasp species. (**A**) *Agelaia myrmecophila*, (**B**) *Polybia sericea*, (**C**) *Polistes major castaneocolor*, (**D**) *Vespula vulgaris*, (**E**) *Stictia* sp., (**F**) *Dasymutilla klugii*. The x-axis is time (minutes); the y-axis is relative intensity (0–100%). Reconstructed mass in Daltons is shown above each peak.

**Figure 7 toxins-15-00224-f007:**
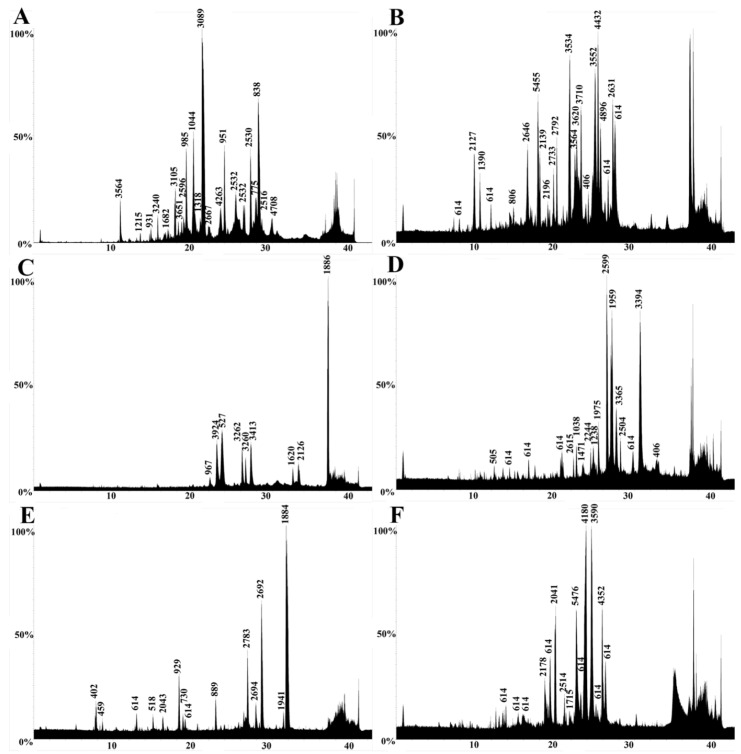
Representative LC-MS profiles of Formicidae species. (**A**) *Dinoponera gigantea*, (**B**) *Myrmecia rufinodis*, (**C**) *Pachycondyla crassinoda*, (**D**) *Platythyrea strigulosa*, (**E**) *Paltothyreus tarsatus*, (**F**) *Odontomachus* sp. The x-axis is time (minutes); the y-axis is relative intensity (0–100%). Reconstructed mass in Daltons is shown above each peak.

**Figure 8 toxins-15-00224-f008:**
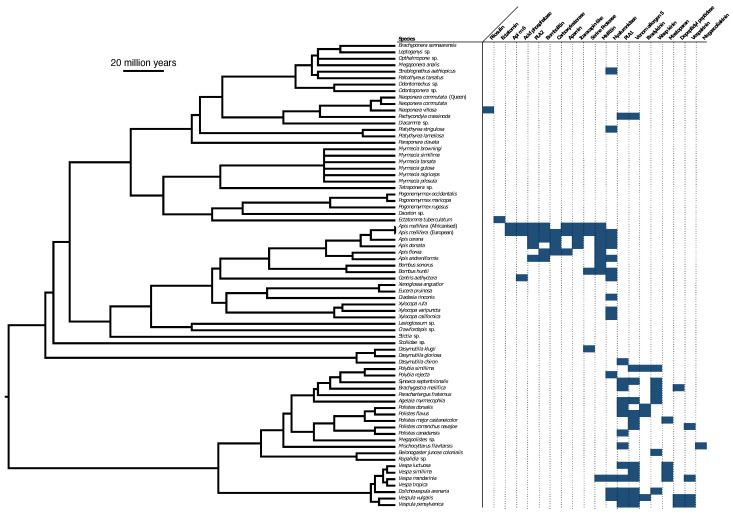
A phylogeny of venom samples which were analyzed though LC-MS/MS and the toxins in the reference database which returned matches to peptides in those venoms. Phylogeny topology and branch lengths from TimeTree (https://timetree.org/) and other previously published phylogenies [81,82,83,84,85,86,87,88,89] were used to manually construct a combined phylogeny in Mesquite 3.7 [90].

**Figure 9 toxins-15-00224-f009:**
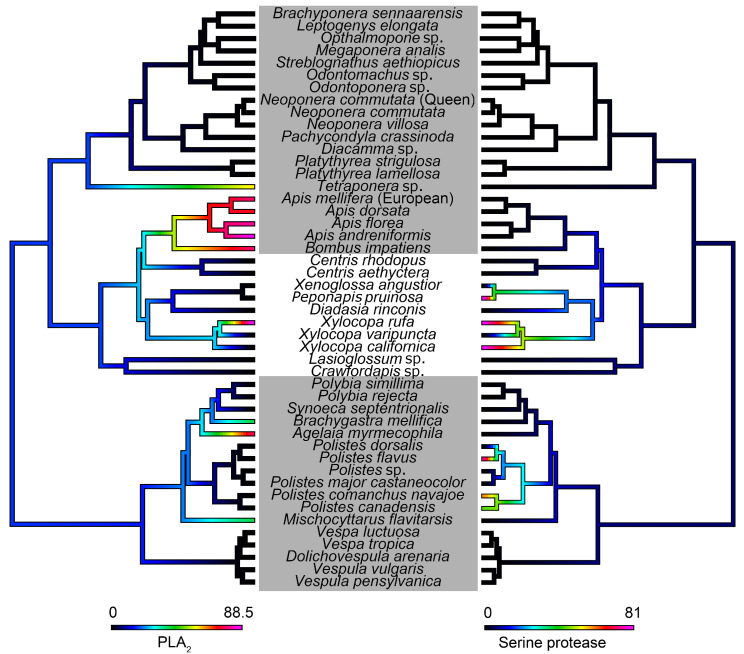
Ancestral state reconstructions of PLA_2_ activity (**left**) and serine protease activity (**right**). Activity was measured as relative percentage absorbance, and warmer colors represent higher activity. Grey boxes indicate social species. Phylogeny topologies and branch lengths from TimeTree (https://timetree.org/) and other previously published phylogenies [81,82,83,84,85,86,87,88,89] were used to manually construct a combined phylogeny in Mesquite 3.7 [90].

**Figure 10 toxins-15-00224-f010:**
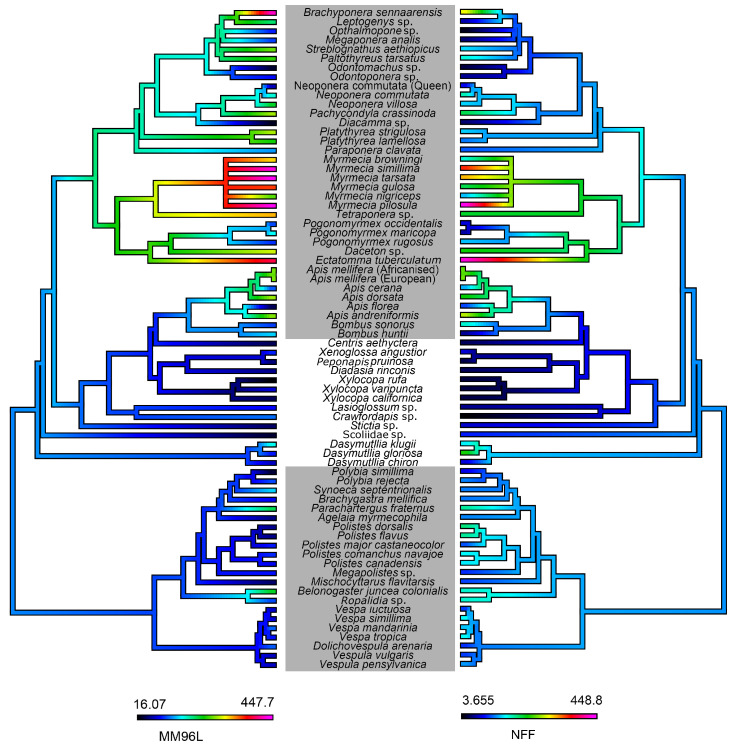
Ancestral state reconstructions of the cytotoxic effects of aculeate venoms against melanoma (MM96L) cancerous cells (**left**) and the non-transformed (NFF) cell line (**right**). Cytotoxicity was measured using the area under the curve of cell mortality over the course of the assay. Warmer colors represent greater toxicity. Grey boxes indicate social species. Phylogeny topologies and branch lengths from TimeTree (https://timetree.org/) and other previously published phylogenies [81,82,83,84,85,86,87,88,89] were used to manually construct a combined phylogeny in Mesquite 3.7 [90].

**Table 1 toxins-15-00224-t001:** Taxonomic sampling of species investigated.

Group	Family	Subfamily	Species
Social Bees	Apidae	Apinae	*Apis andreniformis*
	Apidae	Apinae	*Apis cerana*
	Apidae	Apinae	*Apis dorsata*
	Apidae	Apinae	*Apis florea*
	Apidae	Apinae	*Apis mellifera ligustica* (European)
	Apidae	Apinae	*Apis mellifera scutellata* (Africanised)
	Apidae	Apinae	*Bombus huntii*
	Apidae	Apinae	*Bombus sonorus*
Solitary Bees	Apidae	Apinae	*Centris aethyctera*
	Apidae	Apinae	*Diadasia rinconis*
	Apidae	Apinae	*Xenoglossa angustior*
	Apidae	Xylocopinae	*Xylocopa rufa*
	Apidae	Xylocopinae	*Xylocopa californica*
	Apidae	Xylocopinae	*Xylocopa varipuncta*
	Colletidae	Diphaglossinae	*Crawfordapis* sp.
	Halictidae	Halictinae	*Lasioglossum* sp.
Social Wasps	Vespidae	Polistinae	*Agelaia myrmecophila*
	Vespidae	Polistinae	*Belonogaster juncea colonialis*
	Vespidae	Polistinae	*Brachygastra mellifica*
	Vespidae	Polistinae	*Mischocyttarus flavitarsus*
	Vespidae	Polistinae	*Parachartergus fraternus*
	Vespidae	Polistinae	*Polistes canadensis*
	Vespidae	Polistinae	*Polistes comanchus navajoe*
	Vespidae	Polistinae	*Polistes dorsalis*
	Vespidae	Polistinae	*Polistes flavus*
	Vespidae	Polistinae	*Polistes major castaneocolor*
	Vespidae	Polistinae	*Polybia rejecta*
	Vespidae	Polistinae	*Polybia simillima*
	Vespidae	Polistinae	*Ropalidia* sp.
	Vespidae	Polistinae	*Synoeca septentrionalis*
	Vespidae	Vespinae	*Dolichovespula arenaria*
	Vespidae	Vespinae	*Vespa luctuosa*
	Vespidae	Vespinae	*Vespa mandarinia*
	Vespidae	Vespinae	*Vespa simillima*
	Vespidae	Vespinae	*Vespa tropica*
	Vespidae	Vespinae	*Vespula pensylvanica*
	Vespidae	Vespinae	*Vespula vulgaris*
Solitary Wasps	Mutillidae	Sphaeropthalminae	*Dasymutilla chiron*
	Mutillidae	Sphaeropthalminae	*Dasymutilla gloriosa*
	Mutillidae	Sphaeropthalminae	*Dasymutilla klugii*
	Scoliidae	Scoliinae	Scoliidae sp.
	Crabronidae	Bembicinae	*Stictia* sp.
Ants	Formicidae	Ectatomminae	*Ectatomma tuberculatum*
	Formicidae	Mymicinae	*Pogonomyrmex maricopa*
	Formicidae	Mymicinae	*Pogonomyrmex occidentalis*
	Formicidae	Mymicinae	*Pogonomyrmex rugosus*
	Formicidae	Myrmeciinae	*Myrmecia browningi*
	Formicidae	Myrmeciinae	*Myrmecia gulosa*
	Formicidae	Myrmeciinae	*Myrmecia nigriceps*
	Formicidae	Myrmeciinae	*Myrmecia pilosula*
	Formicidae	Myrmeciinae	*Myrmecia simillima*
	Formicidae	Myrmeciinae	*Myrmecia tarsata*
	Formicidae	Myrmicinae	*Daceton* sp.
	Formicidae	Paraponerinae	*Paraponera clavata*
	Formicidae	Ponerinae	*Brachyponera sennaarensis*
	Formicidae	Ponerinae	*Diacamma* sp.
	Formicidae	Ponerinae	*Leptogenys* sp.
	Formicidae	Ponerinae	*Neoponera commutata*
	Formicidae	Ponerinae	*Neoponera commutata* (Queen)
	Formicidae	Ponerinae	*Neoponera villosa*
	Formicidae	Ponerinae	*Odontomachus* sp.
	Formicidae	Ponerinae	*Opthalmopone* sp.
	Formicidae	Ponerinae	*Megaponera analis*
	Formicidae	Ponerinae	*Pachycondyla crassinoda*
	Formicidae	Ponerinae	*Paltothyreus tarsatus*
	Formicidae	Ponerinae	*Platythyrea lamellosa*
	Formicidae	Ponerinae	*Platythyrea strigulosa*
	Formicidae	Ponerinae	*Streblognathus aethiopicus*
	Formicidae	Ponerinae	*Tetraponera* sp.

## Data Availability

Transcriptomic data has been uploaded to GenBank and are available under the following accession numbers: reads in SRA at SRR11349374, BioProject PRJNA613391, annotated CDS sequences OM416840-OM41685. Raw mass spectrometry results have been uploaded to MassIVE under the accession number MSV000091399.

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
