# Peer review of "Functional and Proteomic Insights into Aculeata Venoms"

_toxins, 2023, doi:10.3390/toxins15030224_

Round 1

Reviewer 1 Report

The manuscript entitled “Functional and proteomic insights into Aculeata venoms: Evolutionary and toxinological implications” investigates the venom composition of different aculeate hymenopterans based on functional and compositional (venomics) experiments. In particular, the venom of honey bees (A. mellifera) is investigated by proteo-transcriptomics. The manuscript is  interesting and provides some novel insights into the venoms of this important class of venomous arthropods. The methods are adequate to address the questions asked and the analysed dataset is impressive as it represents the largest dataset of hymenopteran venoms analysed so far. I consider the manuscript as  relevant to the audience of Toxins.

However, while the scientific rigor of the current draft is excellent, the manuscript requires quite extensive modification. Across the draft, I found several typos that need to be corrected. Also, in the current form the manuscript is very difficult to understand and some aspects of the material and methods part are difficult to reproduce, thus need to be optimized before the manuscript can be published.  Below I provide a detailed list of issues that need to be optimized prior publication.

Line 6: Proteomic data is based on mass spectrometry. Please rephrase for congruency.

Line 7: „She“, I guess it should be “shed”?

Line 8: Exchange “differing” with “different”.

Line 19: Its either most or every, but not both.

Line 42: PLA2 has not been introduced as an abbreviation for phospholipase A2 before.

Section 2 (General): According to the journal guidelines, the Mat/Met part has to follow the major manuscript body. Please modify according to the journal guidelines.

Section 2.1: This section does not give sufficient detail to reproduce the results. Please define precisely which species have been milked via the Schmidt protocol and which have been dissected. Which settings were used in the Schmidt protocol and on which process is it based (is it electrostimulation?). For dissections, please state how the dissected venom glands were treated and which solvents have been used. Did you wash the glands prior to opening them? I also suggest to remove the venom collection part from the taxonomic selection and place it into a separate paragraph.

Section 2.2.1: Again, please add here the details instead of just referring to other works.

Section 3: This section demands more work. Foremost, in its current form it is written in a very complicated way and “jumps” from a study focusing on honey bees straight to other taxa without any clear connection or introduction. Keeping track of the manuscripts storyline requires extensive effort. This stems partly from the missing sub-paragraphs. For instance, there is a section “3.1 Transcriptome” but no other sub sections in the results part. As a result, the transcriptome part houses the proteo-transcriptomic analysis of A. mellifera, the gel analyses, the mass spec results and also the bioactivity/ ancestral state data. The section must be sorted, cleaned and explained clearer prior publication.  Also, the introduction stated that a proteo-transcriptomic analysis of honey bee venom was conducted. If so, it would be important to tell us here what the results of this experiment were. Currently only sequence alignments of identified toxins are shown, but it would be interesting to see what the venom composition of the authors animals look like. A pie chart showing the venom composition, both from protein diversity and abundance, would be very desireable here.

Figure 2: Please add an “of” behind SDS-PAGE.

Figure 3: The gel photographed in panel C is not straight and thus difficult to interpret. I suggest the authors to re-do the electrophoresis.

Table 2: I think a table is not a good way to summarize the venom components from all these taxa. Instead, I would use a figure (maybe based on a phylogeny) and plot the occurrence of components into it. In the current draft, it requires extensive searching to find a species for a toxin.

Line 226-229: I would de-emphasize this a bit as, over the last few years in particular, quite a few studies incorporating complete proteomic plus transcriptomic surveys, were published. I recommend the authors to read these works and implement their findings. For instance von Reumont and colleagues investigated the venom profile of Xylocopa violacea (10.1016/j.toxcx.2022.100117), a solitary bee. Özbek et al studied the venom from Pimpla turionella via proteomics and transcriptomics (10.3390/toxins11120721). Hurka et al published proteo-transcriptomic datasets of two myrmicine and species (10.3390/toxins14050358) and a recent work from the Touchard group even published proteo-transcriptomic data from six myrmicines (10.1016/j.ibmb.2022.103876).

Reviewer 2 Report

1)      L.31 – “Solitary and parasitic species of Aculeata use their venoms in order to paralyse and preserve their prey [2326],” – this is incorrect. Large part of solitary Aculeata are bees, and they do not paralyse and preserve any prey, they feed on nectar and pollen.

2)      L.43 – “The venom from species of Vespidae and Formicidae have also received attention, mostly due to their ability to cause allergic reactions in humans [31,4144].” – the text should connect the study compounds and the allergenic effects of hymenopteran venoms. The link between basophils and venom compounds is not established here at all. The existence of the basophil activation test is also not mentioned. Check and cite, for example, Clin Exp Allergy. 2019 Jan;49(1):54-67, PLoS One. 2015 Nov 12;10(11):e0142953, Mol Immunol. 2022 Sep;149:59-65, and Allergy. 2006 Sep;61(9):1084-5.

3)      Table 1 – “A. mellifera (African and European)” – indicate the subspecies.

4)      Table 1 – “C. rhodipus” – all species names should be spelled in full when mentioned for the first time in the manuscript. It applied probably to all names mentioned in Table 1.

5)      L.97 – “Trimmomatic v0.35 [72] to remove adapter sequences and low-quality reads. Window function-based quality trimming was performed using a window size of 4 and a window quality of 20 and sequences with a resulting length of <100 bp after trimming were removed.” – more details are needed, and probably more stringent settings should be applied.

6)      L.113 – “E10217” – state the linear range of the assay used

7)      L.139 – “fetal calf serum (FCS), 10% FCS” – was it heat-inactivated? And what were the endotoxin levels?

8)      L.144 – “Cell viability was evaluated using colorimetric MTT” – MTT assay may give false-negative results when the metabolism of cells is inhibited despite the cells are still alive. Cell counting, using Trypan blue exclusion, would be the method of choice.

9)      L.152 – “Two independent experiments were conducted with a minimum of three replicates per treatment.” – the number of experiments performed is insufficient. A minimum of three independent experiments should be performed.

10)   Fig. 7 – “sp” should not be italicized.

11)   Fig. 7 –“Phylogeny topology and branch lengths were manually assembled using TimeTree in conjunction with previously published phylogenies [7987].” – the methods used to estimate the tree are undisclosed and unclear. More details are needed.

12)   Fig. 8 – “Ancestral state reconstructions the melanoma (MM96L) cancer line (left) and the NFF cell line.” – this figure legend is completely unclear.

13)   L.218 – “both healthy and cancerous cell lines” – the term “healthy” is usually not used in this context.

14)   L.256 – “Solitary wasp venoms are most likely rich in proteins that are used in order to kill and immobilise prey [109], while solitary bee venoms most likely are high in antimicrobial peptides [4549,110,111].” – state clearly why the solitary bees, but not solitary wasps, should have venoms rich in antimicrobial peptides. Were wasp venoms tested for their presence?

15)   L.276 – “PLA2 and serine protease are also major allergens found in aculeate venoms [32, 33,115]. PLA2 is known to be the main enzyme component found in honeybee venoms, making up approximately 12% of the dry weight of venom [116,117]. Comparatively, wasp venoms have been found to only have 0.1-1% of the protein present [51]” – despite that, allergies to Vespula or Polistes are common in regions of their occurrence. Therefore, it is not PLA2, which is needed to develop an allergic reaction. It is only one of the options which is characteristic of bee venom allergy.

16)   Ref. #10 – volume is lacking.

17)   Ref. #22 – article number should not be provided in the form of a page range

18)   Ref. #73 – “644–52” – formatting of references is inconsistent – here, for example, is the format of page numbers different from ref. #74, etc.

19)   Ref. #111 – “Lasioglossum laticeps” – use italics where appropriate.

20)   After reading through up to the end of the manuscript, I feel that this manuscript should be treated as a missed opportunity. You analyzed a large spectrum of venoms but failed to provide the “Functional and proteomic insights” promised in the manuscript title. Instead, just some LC-MS spectra and superficial analyses of two well-known proteins present in the serum were provided.

Reviewer 3 Report

Reference:  Manuscript submitted to TOXINS November, 2022 “Functional and proteomic insights into Aculeata venoms: Evolutionary and toxinological implications

General comments:  In this manuscript, the authors are conducting studies on composition, proteomic, transcriptomic and functional analyzes of Aculeate hymenopterans venoms, seeking comparative data between venoms of solitary and social species. Throughout the work, the authors show that although many toxins are conserved in different species, there are also variations between venoms with an emphasis on PLA2 and serine protease, as well as on cytotoxic activities. The authors show that species with social behavior have venoms with a higher concentration of peptides involved in tissue destruction and pain in victims. In conclusion, the data are very interesting and relevant to Toxins' scope and deserve to be published. Attached are some suggestions that may make the publication more attractive to readers in the field and that reinforce the scientific aspects of the manuscript.         

Specific Comments

1-    Undoubtedly, a figure and/or a graph showing the analysis flowchart at the end of the introduction could make the text more attractive and complete. The authors could show photos of more representative species within the subclass Aculeata. This would facilitate the understanding of readers, including with clinical interest.  

2-    The authors wrote between lines 32 to 34 …. A review of the toxins found in Vespid venoms concluded that the social and solitary species of that family express very different toxins from each other [31]. But what about between only social species, or only solitary ones, are there also differences in the composition of the venoms? This could give rise to a theory different from that proposed by the authors, who suggest that venoms are directed towards social functions or functions related to the solitary habits of different species. In solitary species there is also a defense function! What is the opinion of the authors about this?

3-    The authors wrote between lines 46 to 47 …. This is an important aspect, as evolutionary selection pressures ‘see’ the entire venom composition, not individual toxins. This statement has its explanations justified in different studies, but also has questions in other studies, since there are toxins that prevail in some venoms and end up alone being responsible for a large number of biological functions seen in crude venoms. Would you like the authors' opinions on this?

4-    The authors wrote between lines 49 to 50 Many of these reactions have been attributed in part to PLA2 and serine protease enzymes. But they also could include hyaluronidases in this group of venom toxins!

5-    The authors wrote between lines 52 to 53 Recent studies have identified small peptidic toxins from a range of aculeates that also disrupt cell membranes [55–59].  In this case, it seems to me that the sentence is incomplete, as the authors could better classify these toxins, these peptides with low molecular mass, from a biochemical point of view. Would they be knottins (ICKs)?

  6-    Also on possible biotechnological applicability of toxins discussed at the end of the introduction and with emphasis on antitumor activities. In this part the authors could also discuss the eternal search by science for drugs with anti-inflammatory activities, as important as the anti-tumor ones, and which have in different Aculeata venoms important examples for studies.

7-    The authors wrote between lines 70 to 71  … The venom has been predominately collected via dissection of the venom glands, but some were milked as described by Schmidt et al. [68].  Why did the authors not standardize the collection of venoms only by dissection of the venom glands, avoiding criticisms of contamination by egesta content of animals manipulated by other procedures?

8-    The authors wrote between lines 74 to 75  … One-Dimensional (1D) sodium dodecyl sulfate polyacrylamide gel electrophoresis 75 (SDS-PAGE) was carried out as previously described [69–71]. It would be more interesting for the authors to complete the information from the SDS-PAGE carried out during the experimental procedures. Acrylamide concentration, whether a linear gradient was used, reduced conditions or not, and the range of molecular mass markers used!

9-    In the line 77 … LC-MS and HPLC analyses of 25 µg crude venom was performed. Why reason the authors used 25 µg crude venom? Is this concentration enough to detect low concentration toxins in different venoms? 

10- In the line 88  … Apis mellifera specimens were sampled for transcriptomic analysis. More details could be indicated as number of animals used, if the sample respected sex division containing female and male are some suggestions!

11- Also why Apis mellifera was chosen among other representative species?

12- In the line 122 … For testing on RDES substrate. What mean RDES? The meaning of abbreviations can be indicated as they appears at the first time in the text.

13- Lines 134 to 156 Cytotoxicity assays … some specific reason why the authors chose human neonatal foreskin fibroblast (NFF) and malignant melanoma (MM96L) cell lines?  

14- They could perform hemolytic assays, test for endothelial cell activation and inflammatory response, basophilic degradulation assays, among others, since hemolytic, inflammatory and allergic reactions are described in accidents involving animals of this class.

15- Some special reason why extracellular matrix of dermis was not studied since according authors skin is a target for venom toxins?

16- In the cytotoxicity experiments the authors wrote line 146 Venom was added to cells at 5 g and 0.5 g protein amounts. I found these protein concentrations to be very high, would not be micrograms? Please check this?

  17-        Finally, it is very important the authors to detail how many animals were used to prepare a stock solution of venoms, which were used to proteolytic, phospholipase and cytotoxicity assays? This information was not present in text. Venom collected from just one animal does not represent a significant sample!  

18- About figure 1, in the results chapter, the authors could detail more explanations in the caption! What do the different colors mean? Red, blue, different shades of black and gray. Preserved amino acids? Equals? Among other details.

19- Also more details of legend of figure 2 and 3 could be incorporated to the text. SDS-PAGE was performed in a linear gradient? Acrylamide concentration? Dye used to stain the gels?

20- Also taking in account that 10 Kda at the lower limit of gel it may not have been enough to detect smaller peptides like some ICKs normally present in different venoms! Is low molecular mass peptides important for venom toxins descriptions as general?

21- The authors wrote among lines 188 to 192 …. Despite the high venom complexity for some of the solitary species shown by the gels, shotgun-MS/MS analysis was only able to find similar matches to a relative handful of toxins (Table 2). This is likely because there are relatively few published homologous sequences available in public databases for us to search our mass spectra against.  Does the fact that the literature has published only a few sequences of protein toxins found in venoms of Aculeata species make the method inappropriate or unsuitable for the authors' purposes?  

22-  In the lines 194 to 195 and 198…Venoms were also profiled using LC-MS to examine the low molecular weight components. Please substitute low molecular weight by low molecular mass, as molecules do not have weight but instead they have masses! You used mass spectrometry and not weight spectrometry!!!

23- In table 2, the presence of different toxins found mainly in social Aculeata species in comparative analyzes with solitary Aculeata species can serve as data on the phylogenetic diversity of the venoms, or just because these toxins have not yet been studied in the venoms of solitary species . Not having been found yet doesn't mean they don't exist! What do the authors think of it?

24- Figure 7 is very interesting, as it aligns phylogenetic analyzes based on the amino acid sequence conservations of toxins in different social and solitary species and at the same time aligns intensity of activities for phospholipases A2 and serine proteases. What is questionable in this figure is that phospholipases A2 prevail only in social species, since the data also show the presence of activities in solitary species (Box Claro, Xylocopa rufa), or am I mistaken? In addition to most of the studied social species having low activity for PLA2 (upper and lower part of the phyllogram). I Would like the authors' opinions?

25- Also use serine proteases in the plural as there are several enzymes with this classification in the different venoms.

26- The data on serine proteases is very interesting and makes sense, since these enzymes are used to mobilize components of the Extracellular Matrix of prey and their presence in solitary species would be very useful. However, proteolytic enzymes have been described in several venoms where they can act as spreading factors for other toxins and this could also be useful in social species. I Would like the authors' opinion?

27- In the line 210 please complete extracellular matrix.

28- The authors need to explain if the venoms of the species that showed greater activity for Serino proteases were collected by another method other than dissection of glands and rule out contamination by oral egesta during venom collections by milking.

29- The upper portion of the phyllogram figure 8, shows greater cytotoxic activities of venoms on melanoma cells and fibroblasts (they are social animals), but the lower portion of figure 8, shows lower cytotoxic activities of venoms on melanoma cells and fibroblasts (they are also social animals). According to the hypothesis developed by the authors, evolutionary pressure could explain a greater cytotoxic activity for social animals. I Would like the authors' opinion on this fact?

30- Between lines 235 to 237 the authors wrote .... Defensive venoms have frequently been noted to be less variable than predatory venoms, so the purpose of the venom may clarify the extreme similarity of A. mellifera toxins [105–107]. Please better define Defensive venoms and Predatory venoms!

31- At line 265 please substitute … low-weight molecules by low-mass molecules.  

Reviewer 4 Report

In this manuscript there are a number of serious shortcomings that in my opinion prevent the publication of this manuscript in its present form. These shortcomings are mostly related to the methods (or rather the lack of e.g. required details) and interpretation of the results, but are also of a conceptual nature.

Introduction and context: The authors make rather grand claims about the evolutionary (or rather behavioral?) transitions between eusocial and solitary species in the subclade Aculeata, the breadth of which is supposedly well covered by the species sampling the authors have performed. However, even subtle (or actually not so subtle) differences in life history traits within the subsocial species, such as the ones listed by the authors, i.e. feeding style and diet, could result in larger changes in the salivary/venom composition, making it hard to interpret any data, and especially the very limited data available in this manuscript.  

Lines 46ff:  I am not sure if it is the way the sentence is phrased, or whether the authors really meant to make this statement, but this is not correct. What do the authors mean by “evolutionary selection”? Do they actually mean natural selection? And if so, the mechanism of natural selection in organisms is through selectively reproducing changes in its genotype. In the context of the complex venom, it also means that changes in the genotype through mutation but also recombination and gene duplication events can result in the altered regulation and/or generation of genes encoding for proteins with new functions. Such changes in overall venom activity can occur fast and can hamper the interpretation of venom composition the way the authors propose, especially the way it is phrased here, namely that “…evolutionary selection pressures (?) “see” the entire venom composition, not individual toxins…”

Importantly, although the assays used with crude venom (but see my comments regarding venom collection) seem adequate, to screen for components of the venom mixtures the authors have only used SDS-PAGE combined with LC/MS and not MS/MS approaches. This results in only LC-MS spectra (Figures 5 and 6), which no reader will be able to compare in a meaningful way using only the provided figures. More importantly, without a real identification of venom peptide/protein sequences, any attempt at comparing venom composition is extremely limited or impossible. This is also true because many venom protein families can be subject to a highly dynamic birth-death process, with species-specific duplication events. In addition, the acquisition of novel genes through gene recruitment and the evolution of orphan genes cannot be addressed using the dataset provided by the authors. A manuscript claiming to focus on comparative venom composition (“evolutionary trajectories”) between solitary and eusocial species at a larger scale has to provide more data than simply adequate activity assays. Therefore the claims made in the introduction are far too grand and the whole manuscript should be rewritten.

Results: I will only focus on one aspect of the provided results, since most of my focal critique was already listed above. In Figure 1 the authors show partial alignments of amino acid sequences of several proteins from Apis mellifera from NCBI and translated sequences from their own transcriptome assembly. This figure is not informative at all, since what exactly do the authors want to tell us here? That some selected proteins are conserved between different Apis mellifera populations? That there are few differences between the transcriptome assembly obtained from their Apis mellifera population and the one found in NCBI? What is even more puzzling is the fact that the authors ignore that at least carboxylesterase (and in other species also phospholipase A2) are members of larger or at least mid-sized gene family, which usually means that only by including all members of the gene family does one get a true impression of the level of sequence similarity/or divergence. In any case, figure 1 does not provide any information beyond a truly basic and rather trivial level.  

Methods: In general, the methods lack essential details and are far too short to really understand what has been done in which way. For example, for a comparative approach the venom collection method matters, especially when comparing this many species. In Apis it is known that e.g. phospholipase A2 secretion into venom follows a seasonal pattern. Having used different methods (and ages of individuals) for venom collection introduces levels of variability which at least needs to be discussed. Furthermore, the authors have only used SDS-PAGE combined with LC/MS and not MS/MS approaches, and even these limited methods need better descriptions, providing essential information lacking here. Likewise, transcriptomic data was only generated for a single species, namely Apis mellifera, for which numerous other datasets exist. For all other species, no transcriptomes were available, which prevents detailed identification of putative venom proteins.  

Round 2

Reviewer 2 Report

The manuscript was significantly improved. However, some comments were reflected to only partial extent or were not reflected at all. These include comments #2 (the link between individual compounds and allergies needs to be highlighted in line of the comment), #5 (just compare the analysis to that using the more stringent criteria, which would or would not support your claim provided in the rebuttal letter), #6 (it is upon you to perform the calibration from which the linear range can be easily assumed), #8 (just add the supporting data that were obtained using some other method), #9 (three independent experiments are a must), #11 (I did not ask how you have drawn the tree but how did you obtain the data used to draw the tree).

Reviewer 4 Report

The authors have extensively revised their manuscript. I really appreciate the amount of work that went into this revision. Although I did not find all of the authors´ responses to my comments adequate (several of my comments were not really addressed at all or brushed aside), there are certainly always different opinions on what the level of hypothesis-testing, discussion and/or context a manuscript has to provide in order to be published. Having this in mind I would now recommend the acceptance of this manuscript.

Author Response

We thank the reviewer for their time and input on the manuscript. We feel it is significantly improved because of their suggestions and those of the other reviewers.